# Histomorphometric Analysis of the Endometrium of Jennies (*Equus asinus*) and Mares (*Equus caballus*) in Estrus: Anatomical Differences and Possible Reproductive Implications

**DOI:** 10.3390/ani16010143

**Published:** 2026-01-04

**Authors:** Pilar Vallejo-Soto, Jesús Dorado, Rafaela Herrera-García, Carmen Álvarez-Delgado, Jaime Gómez-Laguna, Álvaro de Santiago, María Manrique, Antonio González Ariza, José Manuel León Jurado, Manuel Hidalgo, Isabel Ortiz

**Affiliations:** 1Veterinary Reproduction Group (AGR-275), Department of Animal Medicine and Surgery, Faculty of Veterinary Medicine, Universidad de Córdoba, Ctra. Madrid-Cadiz, Km 396, 14071 Cordoba, Spain; v82vasop@uco.es (P.V.-S.); v22hegar@uco.es (R.H.-G.); mhidalgo@uco.es (M.H.); iortiz@uco.es (I.O.); 2Department of Anatomy and Comparative Pathology and Toxicology, Pathology and Immunology Group (UCO-PIG), UIC Zoonosis y Enfermedades Emergentes ENZOEM, Universidad de Córdoba, International Excellence Agrifood Campus ‘CeiA3’, Ctra. Madrid-Cadiz, km 396, 14071 Cordoba, Spain; v52aldec@uco.es (C.Á.-D.); v92golaj@uco.es (J.G.-L.); 3Military Centre of Equine Breeding of Ecija, C. Nueva, 2, 41400 Ecija, Spain; asanlo1@oc.mde.es (Á.d.S.); mmanriquecobo@gmail.com (M.M.); 4Agropecuary Provincial Centre, Diputación de Córdoba, Ctra. Madrid-Cadiz, Km 396, 14014 Cordoba, Spain; aga07@dipucordoba.es (A.G.A.); jmlj01@dipucordoba.es (J.M.L.J.)

**Keywords:** biopsy, donkey, equine, gland, histomorphometry, horse, reproduction, uterus

## Abstract

The donkey population has significantly decreased over the last few decades, and some breeds are now considered endangered. Assisted reproductive techniques (ARTs) play a key role in saving donkeys from extinction. However, the application of current ARTs often yields poorer fertility results in donkeys than in the species they are usually compared to, i.e., horses. It has been suggested that the key to this disparity might lie in the species-specific uterine response, but the anatomical and physiological differences between the uteri of these species remain unknown. In this study, we measured and compared the uterine cells and glands of reproductively sound donkeys (jennies) and horses (mares). Jennies displayed larger and more widely distributed glands than mares. This study provides the first quantitative characterization of the jenny endometrium, highlighting its differences with mares. These species-specific structural variations may partly underlie the distinct uterine response seen during ARTs.

## 1. Introduction

Donkeys have lost their traditional role in industrialized communities, leading to a drastic reduction in their numbers, particularly in developed regions such as Europe, over the last two decades. As a result, most European donkey breeds have become endangered or extinct [1]. Nowadays, donkeys are assuming new roles in society, such as landscape maintenance, flock protection, donkey milk production, and participation in social and leisure activities [2]. Their endangered status and increasing relevance in society makes it crucial to preserve and enhance their genetic pool [3]. Understanding species-specific reproductive physiology is essential, as uterine architecture directly influences fertility and assisted reproductive technologies (ARTs) success [4,5].

ARTs play a key role in the conservation of endangered species [6]. They are often extrapolated from horses to donkeys, despite obtaining poorer fertility outcomes in this species [7,8,9,10]. Crossbreeding studies have suggested that the lower fertility in donkeys is related to an exacerbated inflammatory response of the endometrium of the jenny [9,11]; however, the underlying mechanism remains unclear [12,13,14,15]. A possible explanation could be the differences in the reproductive physiology and anatomy between jennies and mares. Jennies present distinct reproductive behavior, a longer estrus cycle, and a longer pregnancy duration [16,17,18]. From an anatomical perspective, macroscopic qualitative and morphometric analysis of the reproductive tract have revealed key interspecies differences; jennies present the vulva completely below the pelvic brim, proportionally larger internal genitalia, and a narrower and more tortuous cervix forming a deeper fornix [15,19]. Click or tap here to enter text. Microscopic examination of the uterus reveals additional species-specific differences. Endometrial cytology is used to assess the quantity and type of cells present in the uppermost uterine layer, the luminal epithelium of the endometrium [12,13,20]. Although endometrial cytology is useful for evaluating the endometrial environment and detecting inflammation and/or infection, it has inherent limitations.

Uterine biopsy allows evaluation of the cross-sectional anatomical features of the endometrium. In the equine endometrium two layers can be identified: the luminal epithelium and the lamina propria [4]. The lamina propria contains the uterine glands and can be further divided into the stratum compactum (SC), located directly beneath the luminal epithelium, and the stratum spongiosum (SS) [4]. Although the overall structure of the donkey endometrium resembles that of the mare, notable species-specific features have been described, including the qualitative description of more tortuous uterine glands, distinct patterns of immune cell distribution, a characteristic basal inflammatory infiltrate, and differences in fibrotic remodeling [21,22]. Qualitative analysis of endometrial biopsies is routinely used to assess the endometrial health and fertility prognosis of jennies and mares [5,15]. Uterine glands play a crucial role in early equine pregnancy by providing histotrophic secretions essential for embryonic nutrition [23]. Degenerative changes affecting glandular size and distribution have shown to impair early embryonic development and compromise normal placentation [24]. Despite the diagnostic importance of uterine biopsies [5], jennies endometrial samples are evaluated using the mare as a reference. Nonetheless, to the best of our knowledge, no studies have objectively analyzed possible differences with the endometrial structures of reproductively sound jennies.

Histomorphometry has been extensively used to quantitatively evaluate the uterine features of several species, including variations among individuals [25,26], breeds [27], and species [28]. In mares, endometrial histomorphometry has been used to evaluate the effect of the estrus cycle [29,30]; age and reproductive history [29]; susceptibility to endometritis [31]; intrauterine fluid accumulation [32]; and treatment responses [32,33]. Although uterine structure has been previously described in jennies post-mortem [21], comprehensive histomorphometric data in reproductively sound animals with known reproductive histories are lacking. A quantitative comparative study could shed light on the species-specific anatomical differences that might predispose to poorer reproductive outcomes in jennies.

Thus, the objective of this study was to compare for the first time the histomorphometric characteristics of the endometrium of reproductively sound jennies and mares, including luminal and glandular epithelium height, luminal diameter, glandular area, the number of glands and glandular tissue percentage.

## 2. Materials and Methods

### 2.1. Animals and Breeding Soundness Evaluation

A total of twelve animals were included in this study: six mares (*Equus caballus*), with three Purebred Spanish Horse and three Purebred Arabian Horse, and six Andalusian Donkey Breed jennies (*Equus asinus*). The age of the animals ranged from 3 to 16 years old (Table 1), according to their implanted electronic transponders. Age distribution was comparable between jennies and mares. Animals were housed in two herds: the Military Centre of Equine Breeding in Ecija (37º32′28″ N–5º04′45″ W) and the Agropecuary Provincial Centre of Cordoba (37º54′50.9″ N–4º42′40.4″ W), both in Andalusia, southern Spain. Females were kept in paddocks with water available *ad libitum* and were fed grain, straw, and hay. Prior to their inclusion in the study, a reproductive soundness evaluation was conducted. Parity status and reproductive history were recorded. Females were classified as maiden, if they had never been bred; or foaling, if they had foaled 6–8 months before sampling [34]. All foaling animals were evaluated by transrectal ultrasonography to ensure total uterine involution. Females showing a regular estrus cycle with normal ovulation and the presence of a corpus luteum were considered for reproductive soundness evaluation.

The reproductive soundness evaluation consisted of the assessment of cyclic ovarian activity and the absence of pathological images in the reproductive system evaluated by transrectal ultrasonography (Logiq V2, GE HealthCare, Wuxi, China) equipped with a 3.5–10 MHz transrectal probe (LK760-RS, GE HealthCare, Wuxi, China), endometrial biopsy, and endometrial cytology. Cytology and biopsy samples were taken in estrus, as explained below. The animals were considered reproductively sound if they had no previous history of reproductive pathology, showed a previous regular estrous cycle, no free fluid in the uterine lumen, the percentage of polymorphonuclear leukocytes (PMNs) in endometrial cytology smear was lower than 5% [35] and the biopsy was Category I or IIa according to Kenney and Doig [5] endometrial biopsy grading system (KDGS).

### 2.2. Sample Collection

Ovarian cycle control was performed weekly by seriated ultrasound examinations. Endometrial samples were collected when behavioral estrus signs were displayed, in the presence of a dominant follicle > 30 mm in diameter, uterine edema > 2 (scale 0–5, being 0 absence of edema and 5 = hyperedema) [36], and soft, relaxed cervix on vaginal palpation.

After perineal disinfection [37,38], guarded endometrial biopsy was collected by employing the double sleeve technique [39,40] using biopsy forceps with a cutting surface of 4 × 15 mm (Equivet, Kruuse, Kerteminde, Denmark) from the uterine body, aiming at the junction with the uterine horn. Tissue biopsies were immediately tapped onto a sterile slide for cytology and then transferred into a test tube containing 15 mL of 4% buffered formaldehyde fixative solution (Qemical, Quality Chemicals S.L., Barcelona, Spain). Fixation time was no longer than 24 h [21].

A total of 12 endometrial biopsies, 6 from jennies and 6 from mares, were routinely processed and sectioned into 4 μm tissue sections and stained with hematoxylin–eosin [31,41]. Blind biopsy grading was performed by a single pathologist, and histomorphometric analysis was conducted by a single operator who knew the species but ignored age and parity status. The parameters luminal (LE) and glandular (GE) epithelium height, glandular lumen diameter (LD), and glandular area (GA) were measured at 400× magnification, whereas the parameters number of glands (#G), and glandular tissue percentage (GT) were measured at 100× magnification. Each parameter was recorded separately for SC and SS (Figure 1A).

### 2.3. Endometrial Histomorphometric Evaluation

The endometrial histomorphometry was evaluated using NIS-Elements Basic Research software version 5.21.03 (Nikon, Tokyo, Japan) on images provided by a camera (DS-Fi3, Nikon, Tokyo, Japan) connected to an inverted microscope (ECLIPSE Ti1-U IVF, Nikon, Tokyo, Japan).

#### 2.3.1. Measurement of Luminal Epithelium Height and Glandular Size Parameters

The height of the luminal epithelium (LE) was measured as the distance from the basement membrane to the apical edge of the epithelial cell [30] at 400× magnification, following the methodology described elsewhere [29,30,32,42] with modifications (Figure 1B). The LE was calculated as the mean from 30 measures of each endometrium [31]. The results were expressed in micrometers.

The height of the glandular epithelium (GE) was also measured as the distance from the basement membrane to the apical border of the epithelial cell at 400× magnification [29,30,32,42] (Figure 1B,C). For each stratum (glandular epithelium in the stratum compactum, GE-SC; and glandular epithelium in the stratum spongiosum, GE-SS), 30 measures were taken [31] in 10 randomly selected fields per stratum. The results were expressed in micrometers.

Glandular lumen diameter (LD) was calculated as the mean of two perpendicular diameters in circular section glands randomly selected [29,31,32,33] from the SC (LD-SC) and from the SS (LD-SS). Measures were taken at 400× magnification (Figure 1B,C). This parameter was measured in all the circular glands found in SC and SS. To standardize the gland selection for the measures, glands were considered circular when the parameter “*circularity*” was higher than 0.5. This threshold was determined empirically based on our own observations. *Circularity* was measured by automatically detecting the object area by the NIS-Br software and calculated as: Circularity = 4 × π × *Area*/*Perimeter*^2^. The tools *Measure Vertical* and *Measure Horizontal* were used to ensure exact 90-degree perpendicular measures (Figure 1C). The results were expressed in micrometers.

Glandular area (GA) was calculated as the mean area of the glands measured separately for each stratum (glandular area in the stratum compactum, GA-SC; and glandular area in the stratum spongiosum, GA-SS) at 400× magnification (Figure 1D–F). GA was individually quantified using the feature *Automated Measurement* > *Auto detect*, highlighting the selected areas with *Object Colors* as shown in Figure 1D. This tool is based on traditional computer vision algorithms, analyzing pixel intensity levels and variations in adjacent pixels to detect cell or gland boundaries. Automated detection was manually adjusted when histological artifacts or stromal edema compromised detection accuracy. The results were expressed as µm^2^.

#### 2.3.2. Measurement of Glandular Density Parameters

The number of glands (#G) is a glandular density parameter that refers to the number of glands present in a microscopic field. It was measured separately for the SC and the SS (#G-SC, #G-SS), calculated as the mean of the number of glands in 10 fields per stratum at 100× magnification. Glands were quantified as described in previous studies [27,30,42], with some modifications; using the *Object Count* tool of NIS-BR software, disregarding the glands that were touching the image borders. Glands were recognized as individual structures when, regardless of their shape, their structure was delimited by glandular epithelium throughout their perimeter, and there was identifiable stromal tissue around them. The results were expressed as number of glands per field.

Glandular tissue percentage (GT) is a glandular density parameter representing the percentage of the endometrial tissue surface covered by glands. In this study, this parameter was evaluated using the methodology described in cattle by Wang et al. [26] with some modifications. In brief, the area of the glands in the SC and SS was measured separately in 10 microscopic fields, and the corresponding endometrial tissue area was measured using the manual area tool (dashed lines in Figure 1E,F). GT was recorded for each microscopic field, stratum, and sample (GT-SC, GT-SS).$$\mathrm{Glandular}\mathrm{tissue}\mathrm{percentage}=\frac{\sum \mathrm{gland}\mathrm{area}\;({µm}^{2})}{\mathrm{endometrial}\mathrm{tissue}\mathrm{area}\;({µm}^{2})}\times 100$$

**Figure 1 animals-16-00143-f001:**
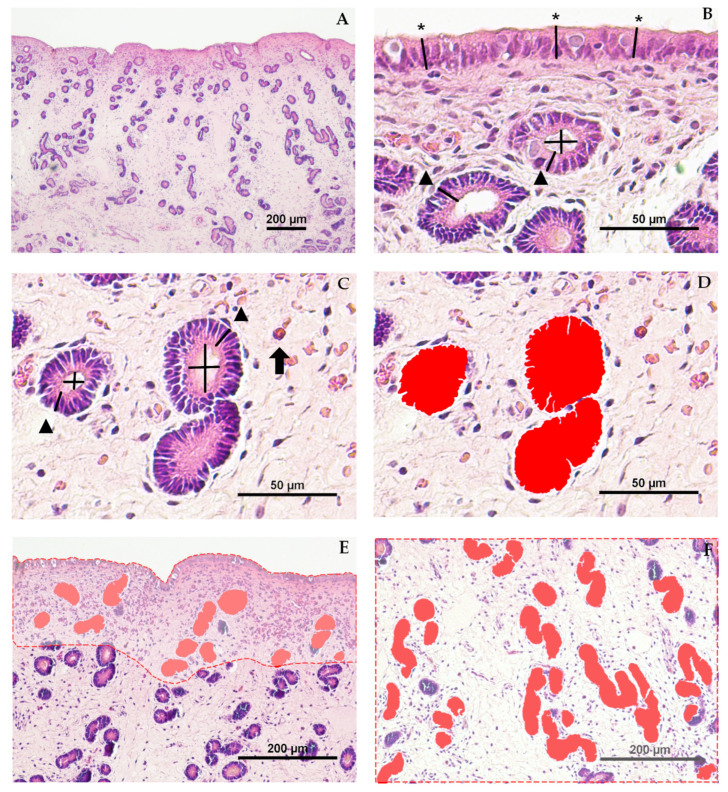
Endometrial histomorphometric evaluation. Hematoxylin and eosin stain. (**A**) Histological section of the equine endometrium (Jenny—J1) stained with hematoxylin and eosin at ×40 magnification; scale bar = 200 µm. (**B**) Example of three measurements for luminal epithelium height (*, asterisk) at ×400 magnification; scale bar = 50 µm. (**C**) Example of two measurements of glandular epithelium height (▲, triangle), and perpendicular luminal diameters (+lines), and an eosinophil (⮕, arrow) at ×400 magnification; scale bar = 50 µm. (**D**) Automated measurement of glandular area, highlighted in red, at ×400 magnification; scale bar = 50 µm. (**E**,**F**) Representative images of the stratum compactum (**E**) and stratum spongiosum (**F**) at 100× magnification for mean glandular area and glandular tissue percentage analysis; scale bar = 200 µm. Glandular areas are shown in solid red, and total endometrial tissue area is outlined with a dashed red line.

### 2.4. Statistical Analysis

All analyses were performed using the statistical package SPSS v29.0.1.0 (IBM Spain, Madrid, Spain). All histomorphometric parameters were compared between species (jenny vs. mare), parity status (maiden vs. foaling), and strata (SC vs. SS). Data distribution normality was assessed using the Kolmogorov–Smirnov test applied to the pooled measurements (30 measurements per animal for size parameters at ×40 magnification; 10 fields per animal for area parameters at ×10 magnification) for each parameter. Due to the limited number of biological replicates, a One-way ANOVA test was used to compare focus on interspecific differences. Post-hoc comparisons were conducted using the Tukey’s HSD test. Results were presented as mean ± SEM. The level of significance was set at *p* < 0.05.

## 3. Results

### 3.1. Breeding Soundness Evaluation

The results of the individual breeding soundness evaluation of the animals included in the study are displayed in Table 1. Histopathological evaluation revealed no pathological fibrosis or inflammatory changes in any of the animals included in the study.

### 3.2. Histological Characteristics of the Endometrial Tissue

The histological characteristics of the endometrial tissue of mares and jennies are illustrated in Figure 2. The luminal epithelium was observed under bright field microscopy (Figure 2A,D), displaying simple cuboidal to columnar epithelium with varying pseudostratification in both species. The lamina propria exhibited estrus-induced stromal edema, and uterine glands showed variable tortuosity among individuals (Figure 2B,C,E,F), which was qualitatively more evident in jennies. Mild subepithelial inflammatory infiltrate was present in all animals. Similarly to cytology results, eosinophils were more frequently observed in jennies than in mares.

### 3.3. Endometrial Histomorphometric Evaluation Results

#### 3.3.1. Results of Luminal Epithelium and Endometrial Glandular Size Parameters

The height of the luminal epithelium (LE), glandular lumen diameter (LD) and glandular area (GA) showed significantly higher values in the stratum compactum (SC) in jennies than in mares (*p* < 0.05, *p* < 0.001 and *p* < 0.05, respectively) (Figure 3). Significant intraspecies differences (*p* < 0.01) between strata (SC vs. SS) were observed in certain parameters: in jennies, GE-SC and LD-SC were larger than their equivalents in SS; whereas in mares GA-SC was smaller than GA-SS.

The differences among the glandular size parameters grouped according to species and parity status were also compared (Table 2). In both species, foaling animals presented higher values (*p* < 0.001) for LE, GE and GA-SC than maiden. Maiden and foaling jennies exhibited wider (*p* < 0.001) LD-SC than mares. There were also significant differences (*p* < 0.01) between strata (SC vs. SS) for some parameters: in maiden jennies, GE-SC was taller than GE-SS; in foaling jennies, LD-SC was wider than LD-SS. Foaling mares exhibited narrower LD-SC than LD-SS; maiden mares had smaller GA-SC than GA-SS.

#### 3.3.2. Results of Glandular Density Parameters

Glandular density parameters revealed further significant differences in glandular tissue distribution between species (Figure 4). The quantification of glands (#G) was similar in both species (*p* > 0.05), while glandular tissue percentage in the SC (GT-SC) was significantly higher in jennies than in mares (*p* < 0.01). In mares, GT-SS proportionally covered a greater area (*p* < 0.05) than GT-SC.

The comparison of density parameters grouped within species and parity status was also evaluated (Table 3). The number of glands (#G) did not significantly differ (*p* > 0.05) among species or parity status. In foaling jennies, GT-SC covered a larger (*p* < 0.001) area compared to mares (both foaling and maiden). Additionally, GT-SC in maiden jennies was significantly larger (*p* < 0.001) than in their mare counterparts. Regarding differences observed between strata, in both species, there were fewer #G (*p* < 0.001) in the SC. No differences between strata were observed for GT (*p* > 0.05).

Although #G did not show significant differences (*p* > 0.05) among species or parity status, the larger glandular area in jennies contributed to an increase in the percentage of glandular tissue (GT) despite showing similar #G in the SC when compared to mares. Accordingly to glandular size parameters, the percentage of glandular tissue covered a greater area (*p* < 0.01) of the SC of the endometrium in jennies than in mares when comparing species within the same parity status.

## 4. Discussion

This study presents the first histomorphometric comparison between the endometrium of jennies and mares, providing objective differences among the endometrial features of these species. This novel characterization of the endometrial tissue of jennies provides valuable insights into their unique physiological traits that could play a role in the different response after artificial insemination in these species [9,11].

Our study revealed significant differences between jenny and mare glandular size parameters. During estrus, the luminal epithelium (LE) was significantly taller in jennies than in mares. The glandular epithelium (GE) of the glands in the stratum compactum (SC) was also taller in maiden jennies than in maiden mares. The observed increase in epithelial height (luminal and glandular) in jennies compared to mares suggests a heightened level of glandular activity [4,43]. This fact was supported by the finding that LE and GE were also taller in animals that had foaled that year than in maiden animals in both species. Glandular lumen diameter and area (LD-SC and GA-SC) were also larger in jennies compared to mares, reinforcing the idea of increased glandular activity. In mares, similar histomorphometric features have been associated with enhanced secretory activity and impaired uterine clearance [32]. Although glandular secretion was not assessed in the current study, exuberant secretion has been proposed to promote glandular enlargement and intraluminal fluid accumulation. This accumulation may trap inflammatory debris and promote further mucus production and bacterial growth, creating a self-sustaining inflammatory cycle [44].

The presence of larger glands in jennies and foaling animals, as indicated by increased LD-SC and GA-SC, may result in an environment rich in uterine secretion and inflammatory mediators, which could facilitate bacterial colonization and increase the risk of uterine contamination [45]. Previous studies have shown that pathogenic microorganisms causing endometritis in mares are able to reside in the epithelial cells and within the endometrial glands [46,47,48,49]. Their presence in these locations has been proposed to reflect distinct survival mechanisms: dormant bacteria inside the cells may evade detection until activated by specific triggers [46]; and within the glandular lumen, phagocytic binding may be impaired [49] while accessing the nutrient-rich glandular fluid [47]. Previous studies have shown that larger glandular structures are more prevalent in older mares with poor reproductive outcomes [29], and larger glands are also related to greater susceptibility to a more exacerbated post-breeding inflammation [31,32]. Similar patterns have been observed in other species, such as the dog; bitches with cystic endometrial hyperplasia show glandular enlargement and mucus retention, creating favorable conditions for bacterial growth and predisposing these animals to suffer from pyometra [50]. It is important to note, however, that the present study included only reproductively sound jennies and mares, and the larger glandular dimensions observed in jennies represent normal species-level anatomical variation. Within this physiological context, wider glands in estrus (as found in foaling females and jennies in particular) could contribute to heightened inflammatory responses, although this hypothesis requires further functional validation through inflammatory challenge and microbial assessments. This hypothesis may help explain some reproductive differences found in jennies; including higher basal polymorphonuclear leukocyte (PMNs) counts in endometrial samples [15,51,52], a more pronounced post-breeding inflammatory response [14,53], and lower pregnancy rates after AI [9,11,54].

In addition to the size of luminal epithelium and glandular structures, glandular density across the stromal layers shows a wider picture of the endometrium. In our study, glandular density was assessed by counting the number of glands per field (#G) and calculating the percentage of tissue covered by glands (GT). The latter incorporated the parameter gland area using a novel methodology for mares, adapted from what has been applied in other species [26]. This approach was chosen for its accuracy and ease of implementation in comparison to previous analogue approaches [27,29,30,31,42], although formal inter- and intra-observer repeatability assessment would further strengthen the robustness of the method. Our findings revealed that, in jennies, GT covered a larger area within the SC, while the #G did not differ significantly between species. The fact that GT covered a wider area in jennies and #G was similar for both species, stands by the finding that jennies display wider glands which accumulate more secretion in the glandular lumen, which may contribute to bacteria colonization and heightened inflammatory response, as previously discussed [44,46,49]. Previous studies conducted in mares (Appendix A) found that an increased number of glands in mares is associated with a higher susceptibility to endometritis [31], while higher glandular density has been linked to intrauterine fluid accumulation [32]. These findings reaffirm that the higher amount of glandular tissue found in jennies might be linked to a more exacerbated post-breeding inflammation. In contrast, a study evaluating the histomorphometric features of older mares with poor reproductive outcomes reported a lower number of glands, concluding that insufficient functional tissue could lead to early embryo death [29]. However, this study did not take into consideration the percentage of glandular tissue these glands represented. These findings emphasize the need for future studies to further investigate glandular density and its potential implications in subfertile mares and jennies.

Remarkably, our results showed that significant differences in glandular size and density between jennies and mares were almost exclusively limited to the stratum compactum (SC), with minimal variations found in the deeper stratum spongiosum (SS). The SC, located directly beneath the luminal epithelium, seems to be more sensitive to hormonal and environmental stimuli. In the SC, histological changes occur more rapidly under the hormonal changes during estrus [4], supported by the increased mitotic activity found in this stratum [55]. In addition, stratal layer differentiation are considered in KDGS, with the percentage of PMNs in the SC widely used as a standard method for identifying endometritis [56,57]. These results highlight the importance of differentiating between strata when performing histomorphometric analyses, as significant differences may be overlooked without this separation.

In order to identify the physiological interspecific differences in the endometrium, this study only included reproductively healthy jennies and mares, meeting the criteria of (i) no endometrial inflammation (PMNs < 5%) and (ii) low degree of fibrotic endometrial degeneration with scores I or IIa according to the KDGS [5]. The relatively small but standard histomorphometric sample size warrants cautious generalization until confirmed by larger studies. Future studies including more animals with reproductive pathologies such as inflammation or fibrotic endometrial degeneration (KDGS IIb or III) could validate these findings and deepen our understanding of how endometrial status affects reproductive adaptations and outcomes in these closely related species.

## 5. Conclusions

The comparison of histomorphometric characteristics between reproductively sound jennies and mares revealed significant endometrial differences, with jennies displaying larger luminal epithelium, glandular size, and density compared to mares. These anatomical differences might play a role in the distinct uterine response after AI in jennies. Further research is needed to understand the physiological mechanism underlying the uterine response in jennies.

## Figures and Tables

**Figure 2 animals-16-00143-f002:**
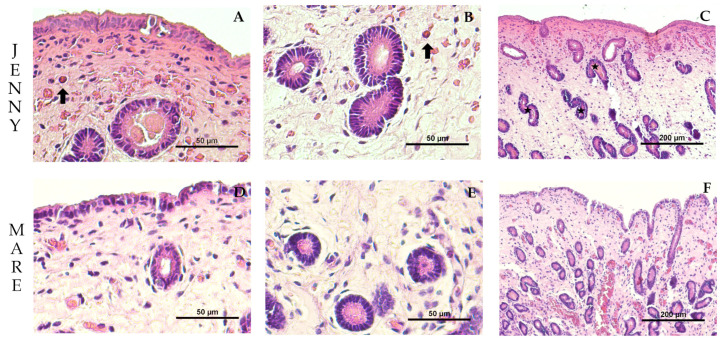
Representative images of the endometrial tissue of a maiden jenny (**A**–**C**) and maiden mare (**D**–**F**) used for histomorphometric analysis. Hematoxylin and eosin stain; (**A**,**B**,**D**,**E**): at 400× magnification, scale bar = 50 µm. (**C**,**F**): at 100× magnification, scale bar = 200 µm. Eosinophils (⮕, arrow); examples of gland tortuosity (⋆, star).

**Figure 3 animals-16-00143-f003:**
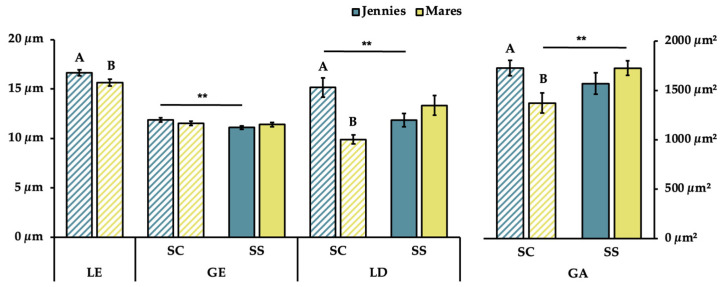
A bar chart representing the mean ± SEM of histomorphometric measurements of luminal epithelium height and glandular size parameters compared between reproductively sound jennies and mares. LE = Luminal epithelium height (µm); GE = glandular epithelium height (µm); LD = glandular luminal diameter (µm); GA = glandular area (µm^2^); SC = stratum compactum; SS = stratum spongiosum. Different capital letters indicate significant differences between jennies and mares for each stratum (A, B *p* < 0.05). The lack of superscripts indicates no significant differences between species (*p* > 0.05). Asterisks indicate significant differences between strata within each species (SC vs. SS; ** *p* < 0.01).

**Figure 4 animals-16-00143-f004:**
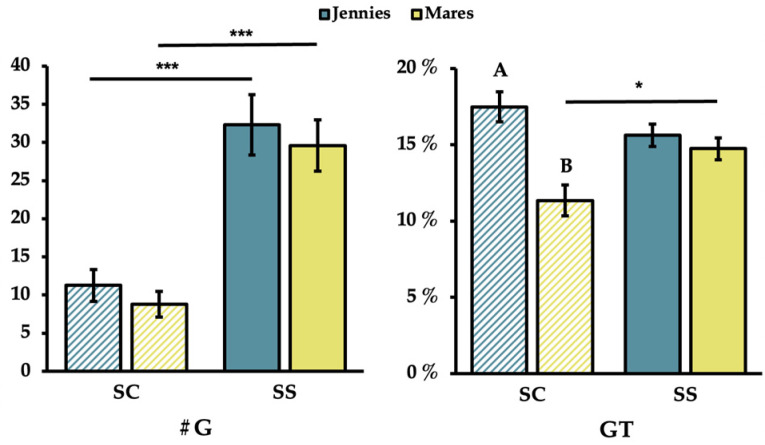
A bar chart representing the mean ± SEM of histomorphometric measurements of glandular density parameters between reproductively sound jennies and mares. #G = number of glands; GT = Glandular tissue percentage. Different superscripts in capital letters indicate significant differences between total jennies and mares for each stratum (A, B *p* < 0.05). The lack of superscripts indicates no significant differences between species (*p* > 0.05). Asterisks indicate difference between strata within species (SC vs. SS): * *p* < 0.05 and *** *p* < 0.001.

**Table 1 animals-16-00143-t001:** The results of the breeding soundness evaluation of the animals included in the study (jennies J1–J6, n = 6; mares M1–M6, n = 6).

Parameters	Jennies						Mares					
	J1	J2	J3	J4	J5	J6	M1	M2	M3	M4	M5	M6
Parity status ^1^	Maiden	Maiden	Foaling	Foaling	Foaling	Foaling	Maiden	Maiden	Foaling	Foaling	Foaling	Foaling
Age (years) ^2^	3	5	10	10	11	12	3	3	12	13	14	16
Breed ^3^	AD	AD	AD	AD	AD	AD	PRE	PRE	Ar	Ar	Ar	PRE
DFD (mm) ^4^	30.5	39	31	30	33	38	36	32	33.5	38.5	31.5	35
Edema ^5^	3	2	4	3	3	2	2	2	3	3	3	4
Cervix ^6^	O	O	O	O	O	O	O	O	O	O	O	O
PMNs (%) ^7^	1.62	0.50	0.17	2.61	3.71	0.66	0.66	0.33	0.50	0.00	0.33	0.00
Neu (%) ^8^	1.35	0.5	0	1.74	2.94	0.33	0.33	0.33	0.50	0	0.33	0.00
Eos (%) ^9^	0.27	0	0.17	0.87	0.77	0.33	0	0	0	0	0	0
KDGS ^10^	I	IIa	I	IIa	I	IIa	I	IIa	IIa	IIa	IIa	I

^1^ Parity status: Maiden: they have not been bred; Foaling, they had foaled 6–8 months before samplingt. ^2^ Age of the animals at the moment of the study. ^3^ Breed: Andalusian donkey breed (AD), Pure Spanish horse (PRE), and Purebred Arabian horse (Ar). ^4^ Dominant follicle diameter (DFD). ^5^ Uterine edema. Scale 0–3 being 0 no edema and 5 maximal edema (hyperedema). ^6^ Cervix: soft relaxed on vaginal palpation (O). ^7^ Percentage of polymorphonuclear leukocytes (PMS, %) in endometrial cytology. ^8^ Percentage of neutrophils (Neu, %) in endometrial cytology. ^9^ Percentage of eosinophils (Eos, %) in endometrial cytology. ^10^ Categories according to Kenney and Doig’s endometrial biopsy grading system (I, IIa, IIb, III).

**Table 2 animals-16-00143-t002:** Descriptive statistical analysis of the histomorphometric measurements of luminal epithelium height and glandular size parameters grouped according to species and parity status.

Parameters	Jennies		Mares	
	Maiden (n = 2)	Foaling (n = 4)	Maiden (n = 2)	Foaling (n = 4)
LE (µm)	14.19 ± 0.47 ^b^	17.88 ± 0.33 ^a^	10.93 ± 0.24 ^c^	18.01 ± 0.37 ^a^
GE-SC	11.23 ± 0.30 ^b^**	12.21 ± 0.26 ^ab^	9.75 ± 0.21 ^c^	12.43 ± 0.25 ^a^
GE-SS	10.10 ± 0.25 ^b^	11.59 ± 0.22 ^a^	9.39 ± 0.23 ^b^	12.43 ± 0.25 ^a^
LD-SC	15.25 ± 1.39 ^a^	15.62 ± 1.27 ^a^**	10.78 ± 0.75 ^b^	9.37 ± 0.55 ^b^
LD-SS	12.80 ± 1.45	11.47 ± 0.71	12.78 ± 1.47	13.71 ± 1.38 **
GA-SC	1322.03 ± 105.85 ^bc^	1908.62 ± 148.64 ^a^	1018.72 ± 74.21 ^c^	1588.38 ± 103.27 ^ab^
GA-SS	1501.21 ± 110.18	1599.21 ± 104.16	1485.8 ± 127.31 **	1835.28 ± 137.12

Parity status = (Maiden: they have not been bred; Foaling, they had foaled the year of the study). LE = Luminal epithelium height (µm); GE = glandular epithelium height (µm); LD = glandular luminal diameter (µm); GA = glandular area (µm^2^); SC = stratum compactum; SS = stratum spongiosum. Values are expressed as mean ± SEM. ^a–c^ Values within a row with different superscripts differ significantly at *p* < 0.05. Asterisks indicate significant differences between strata for each group (SC vs. SS): ** *p* < 0.01.

**Table 3 animals-16-00143-t003:** A descriptive statistical analysis of the histomorphometric measurements of glandular density parameters within species, grouped according to parity status.

Parameters	Jennies		Mares	
	Maiden (n = 2)	Foaling (n = 4)	Maiden (n = 2)	Foaling (n = 4)
#G-SC	11.68 ± 1.51	10.86 ± 1.20	8.70 ± 1.66	8.83 ± 1.22
#G-SS	26.25 ± 3.05 ***	35.53 ± 2.31 ***	31.90 ± 4.24 ***	25.91 ± 2.31 ***
GT-SC	15.67 ± 1.20 ^ab^	18.31 ± 1.33 ^a^	8.70 ± 1.52 ^c^	12.67 ± 1.27 ^bc^
GT-SS	14.37 ± 1.22	16.12 ± 0.91	12.87 ± 0.81 *	15.41 ± 0.99

Parity status = (Maiden: they have not been bred; Foaling, they had a foal the year of the study). #G = number of glands; GT = glandular tissue percentage (%); SC = stratum compactum; SS = stratum spongiosum. Values are expressed as mean ± SEM. ^a–c^ Values within a row with different superscripts differ significantly at *p* < 0.05. Asterisks indicate significant differences between strata (SC vs. SS): * *p* < 0.05, *** *p* < 0.001.

## Data Availability

The data are deposited in HELVIA official repository: http://hdl.handle.net/10396/31093 (accessed on 30 December 2025).

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
