# Peer review of "Histomorphometric Analysis of the Endometrium of Jennies (Equus asinus) and Mares (Equus caballus) in Estrus: Anatomical Differences and Possible Reproductive Implications"

_animals, 2026, doi:10.3390/ani16010143_

Round 1

Reviewer 1 Report

Comments and Suggestions for Authors

Title:
The scientific name of the studied animal must be included, and the specific estrous phase under investigation should also be stated.
Simple summery:
The abstract should be rewritten according to the journal’s required format.
Abstract: 
It must include the importance of the donkey in reproductive and endometrial studies, and the exact number of samples.
Keywords:
Keywords must not repeat the words used in the article title.
Introduction
The introduction and the study objectives require major revision.
A brief description of the endometrial histology of the jenny (female donkey) must be added at the beginning of the introduction.
The following references must be incorporated into this section, and based on their content, the rationale and objectives of the present study should be rewritten:
•    Quartuccio et al., 2020 – Endometrial cytology during different estrous phases in jennies
•    Renner-Martin et al., 2009 – Gross anatomy of female genital organs of the domestic donkey
•    Taberner et al., 2008 – Estrous cycle characteristics and prediction of ovulation in Catalonian jennies

Based on Renner-Martin’s paper, you must explain why further study of the equine endometrium is necessary, since the authors could have compared their current findings in donkeys directly with the equine endometrium data reported in that study.

A description of the common endometrial disorders and pathological conditions in jennies must also be included.
Furthermore, the importance of histomorphometry in such studies must be clearly stated.

Materials and Methods
1.    The method used to determine the age of the animals must be described.
2.    The animals’ body weights must be provided.
3.    The housing and management conditions must be described.
4.    Information on parity, endometrial status, and any uterine pathology must be fully explained.
5.    In line 99, detailed specifications of the ultrasound machine and the corresponding image must be provided.
6.    In lines 106–107, the exact duration of tissue fixation in formalin must be stated.
7.    Lines 175–182: Figure legends must be self-explanatory without needing to refer to the main text. Anatomical terms require correction.
8.    Lines 104–105: The exact sampling sites of the uterus must be described clearly and precisely.
9.    Lines 121–165: Images showing the measured and quantified structures must be presented clearly, even if some are already partially labeled.

Results
10.    Lines 206–207: This part belongs to the Discussion section and should be moved accordingly.
11.    Table titles must be complete, clear, and understandable without referring to the text (e.g., Table 2).
12.    Lines 209–211: Figure legends must be clear and descriptive; relevant tissue structures must be labeled, named, and explained.

Discussion
To facilitate better understanding, a table must be created comparing the findings of the present study with previously published studies on this topic.
The effects of breed, age, and reproductive history on the findings of this study, as well as those reported in the literature, must be discussed.
The study limitations should be expanded and written more comprehensively.

Reviewer 2 Report

Comments and Suggestions for Authors

The manuscript entitled Histomorphometric analysis of the endometrium of jennies and mares: anatomical differences and possible reproductive implications, by authors Vellejo-Soto et al., compares well known methods for breeding evaluation of equine species, to provide answers to poorer outcome of AI techniques in donkeys compared to horses. The paper is nicely written, it is of interest to wider veterinary scientific and professional audience.

But I do have a question and a suggestion for the authors: question regards the timing of uterine samples collection. From the text it is not clear when in reproductive season the sampling was done (probably during the full season in mares?), and how soon after parturition?

The second suggestion: line 206-207: this sentence belongs more to the Discussion.

Reviewer 3 Report

Comments and Suggestions for Authors

Dear Authors

This manuscript provides a valuable and novel quantitative comparison of endometrial histomorphometry between jennies and mares. The study is well-designed, the methodology is rigorous, and the findings contribute meaningfully to the understanding of species-specific reproductive anatomy. However, several sections require clarification, softening of speculative interpretations, and improvements in figure presentation.

I have inserted detailed line-by-line comments directly on the annotated PDF to guide precise corrections.

Section-by-Section Comments

  1. Introduction
  • Clarify the anatomical rationale linking uterine microstructure to ART outcomes.
  • Specify whether "branched glands" reported in jennies in prior work were qualitative or quantitative observations.
  • Strengthen the novelty statement by emphasizing that this is the first quantitative histomorphometric comparison between the two species.
  1. Materials and Methods
  • Indicate the exact biopsy site (uterine horn or uterine body), as gland density can vary regionally.
  • Confirm whether all foaling animals had completed uterine involution.
  • Clarify the timing within estrus, since epithelial height and stromal edema fluctuate.
  • Indicate whether histomorphometric measurements and biopsy grading were performed blinded.
  • Consider a brief justification for using one-way ANOVA instead of a multi-factor model.
  • For small sample sizes, mention whether Shapiro–Wilk was considered for testing normality.
  • Clarify whether image analysis required manual correction when auto-detection failed.
  1. Results
  • Eosinophil prominence in jennies should be identified as a qualitative observation unless quantified.
  • Clarify whether increased gland tortuosity was consistent across all animals.
  • Strengthen explanation that #G does not differ between species, yet GT differs due to larger glandular area in jennies.
  1. Figures
  • Figure 4: The “number of glands per field” panel lacks A/B superscripts. Either add identical superscripts or clearly state in the caption that no significant differences were found.
  • Superscripts, scale bars, and arrow labels should be increased in size for better readability.
  • Consider more distinct color coding to differentiate SC and SS visually.
  • Annotate examples of tortuous glands more clearly in Figure 2.
  1. Discussion
  • Please soften speculative statements linking larger glands or epithelial height to “predisposition to inflammation,” as no inflammatory challenge was studied and all animals were reproductively sound.
  • Clarify that gland enlargement in jennies represents normal species-level variation, not pathology.
  • Provide a brief anatomical explanation for why interspecies differences occur primarily in the stratum compactum.
  • Acknowledge sample size limitations typical of histomorphometric studies.
  1. Abstract & Simple Summary
  • Avoid phrasing that implies jennies have pathological-like features; instead emphasize species differences.
  • Highlight novelty more clearly.
  • Add a phrase acknowledging that GT differences occur mainly in the SC.
  1. Conclusion
  • The conclusion is strong but would benefit from adding that the differences identified reflect normal anatomical physiology rather than pathological susceptibility.

Comments on the Quality of English Language

The English is generally understandable, but several sentences require clearer scientific phrasing, removal of speculative wording, and minor grammatical corrections. A careful language edit is recommended.

Round 2

Reviewer 1 Report

Comments and Suggestions for Authors

“Congratulations”

All comments have been carefully considered and addressed, except for the following comment:  

According to MDPI requirements, the Simple Summary must:

  • Be no longer than 200 words

  • Be written in plain, non-technical language

  • Be understandable to the general public and non-specialists

  • Not repeat or paraphrase the main Abstract

  • Explain the purpose and importance of the study rather than technical details

Author Response

Comment 1: “Congratulations”. All comments have been carefully considered and addressed.

Response 1: Thank you very much. Your valuable comments have greatly improved the quality of this manuscript.

Comments 2: Except for the following comment: According to MDPI requirements, the Simple Summary must:

2.1. Be no longer than 200 words

2.2. Be written in plain, non-technical language

2.3. Be understandable to the general public and non-specialists

2.4. Not repeat or paraphrase the main Abstract

2.5. Explain the purpose and importance of the study rather than technical details

Response 2.1.: The previous version of the simple summary had 198 words. The new version has 144 words.

Response 2.2. and 2.3.: We have thoroughly searched for sentences containing technical terms. We have rephrased them using plain, non-technical language, particularly in lines 23 – 28.

Response 2.4. and 2.5.: We have revised the simple summary to avoid paraphrasing the main abstract. The simple summary now focuses more on the importance of performing this study and the relevance of the obtained results, avoiding technical details.

Reviewer 3 Report

Comments and Suggestions for Authors

The authors have addressed all my comments thoroughly and constructively. The revised manuscript shows clear improvement in clarity, methodological transparency, and interpretation of findings. In particular, speculative statements have been appropriately tempered, the novelty of the quantitative histomorphometric approach is now clearly articulated, and the distinction between species-specific physiological variation and pathology is well maintained.

The study is methodologically sound, the results are clearly presented, and the conclusions are well supported by the data. No further revisions are required.

Author Response

Thank you so much for your kind words and for taking the time to review our work. Your comments have helped us improve the clarity, transparency, and interpretation of findings.